# Remembering is Not Applying: Interpretable Knowledge Tracing for Problem-solving Processes

## ABSTRACT

Knowledge Tracing (KT) is a critical service in distance education, predicting students' future performance based on their responses to learning resources. The reasonable assessment of the knowledge state, along with accurate response prediction, is crucial for KT. However, existing KT methods prioritize fitting results and overlook attention to the problem-solving process. They equate the knowledge students memorize before problem-solving with the knowledge that can be acquired or applied during problem-solving, leading to dramatic fluctuations in knowledge states between mastery and non-mastery, with low interpretability. This paper explores knowledge transformation in problem-solving and proposes an interpretable model, Problem-solving Knowledge Tracing (PSKT). Specifically, we first present a knowledge-centered problem representation that enhances its expression by adjusting problem variability. Then, we meticulously designed a Sequential Neural Network (SNN) with three stages: (1) Before problem-solving, we model students' personalized problem space and simulate their acquisition of problem-related knowledge through a gating mechanism. (2) During problem-solving, we evaluate knowledge application and calculate response with a four-parameter IRT. (3) After problem-solving, we quantify student knowledge internalization and forgetting using an incremental indicator. The SNN, inspired by problem-solving and constructivist learning theories, is an interpretable model that attributes learner performance to subjective problems (difficulty, discrimination), objective knowledge (knowledge acquisition and application), and behavior (guessing and slipping). Finally, extensive experimental results demonstrate that PSKT has certain advantages in predicting accuracy, assessing knowledge states reasonably, and explaining the learning process.

## CCS CONCEPTS

• **Applied computing → Education**; **Distance learning**.

## KEYWORDS

distance education, user modeling, knowledge tracing, problem solving

*ACM MM, 2024, Melbourne, Australia*
© 2024 Copyright held by the owner/author(s). Publication rights licensed to ACM.
ACM ISBN 978-x-xxxx-xxxx-x/YY/MM
https://doi.org/10.1145/nnnnnnn.nnnnnnn

## 1 INTRODUCTION

Digital multimedia technology enables the creation and delivery of high-quality educational content in various digital formats, promoting distance education's popularization. Consequently, online teaching modes represented by MOOCs, micro-courses, and blended learning have emerged [12, 13]. The phenomenon of knowledge reversal occurs when instructional procedures effective for beginners are ineffective for experts [17]. Therefore, educators must be able to continuously assess and monitor learners' knowledge mastery during the learning process to dynamically determine appropriate teaching resources, especially in online learning scenarios.

Knowledge tracing (KT) is an influential research area in distance education, which can automatically track students' knowledge levels at various stages and has been widely used in adaptive learning systems [2, 19]. The goal of KT involves two stages. Firstly, assessing the development of students' knowledge states according to records, and secondly, predicting their performance on specific problems using the assessed knowledge states. Therefore, KT offers two key applications for intelligent educational services: dynamic adjustment of teaching strategies based on assessment states and personalized resource recommendation based on prediction results.

Early KT models, such as Bayesian Knowledge Tracing (BKT) [6, 44] based on Hidden Markov Models, and Performance Factor Analysis [30] based on logistic functions, used educational parameters to estimate students' mastery probabilities. These models were characterized by high interpretability but low precision in prediction. As smart education progresses, deep learning-based approaches (DLKT) have become mainstream, significantly improving the performance of KT tasks [11, 26, 31, 45]. However, due to the black-box nature of deep learning, these models have lower transparency and interpretability. Similar to how good teachers need to understand students, a good KT model should provide higher-value educational services, such as attribution analysis, which helps understand the 'why' - not just the 'what'.

Fortunately, many DLKT methods are addressing this issue by striving to enhance the interpretability of their models. Methods like Deep-IRT [42] and Deep-IRTw [38] combine KT with Item Response Theory (IRT). SAKT [28], AKT [11] utilize the weights of the attention mechanism to explain the relationship between historical interactions. IKT [23] sets knowledge mastery, learning transfer ability, and problem difficulty as three educationally meaningful parameters. RCKT[7], CMKT [47] enhance model interpretability through causal inference.

However, the aforementioned methods primarily focus on prediction and overlook the consideration of the learning process, resulting in an unreasonable assessment of knowledge mastery patterns. LPKT [33] recognized the limitations of previous models, which assumed that learners' knowledge states would decline once they answered incorrectly. LPKT applied knowledge gain to measure the value of error interactions. From a cognitive perspective,

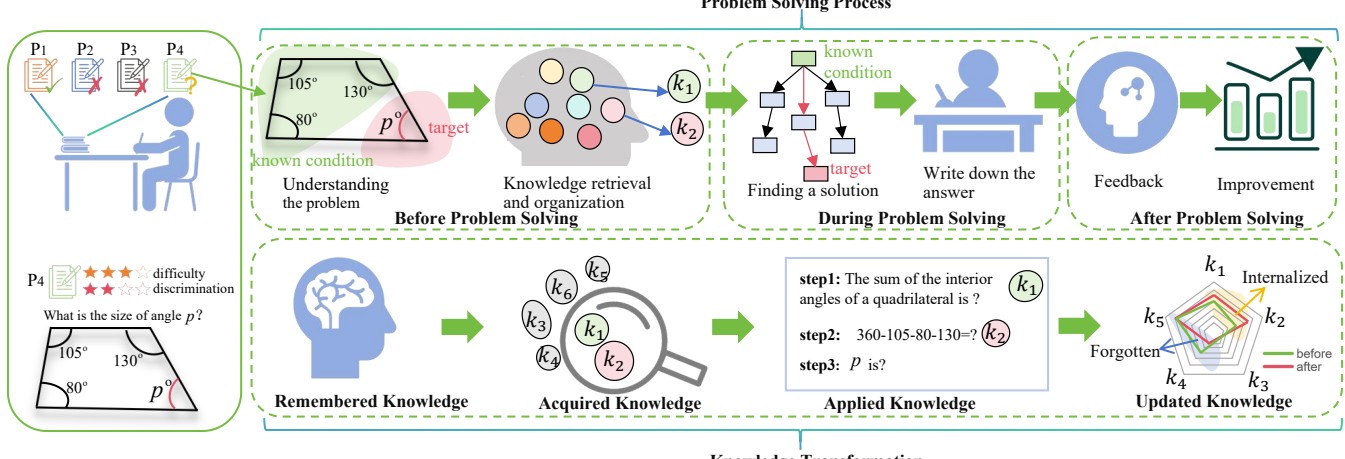

**Figure 1: Profiling the student's problem-solving process. The upper right part illustrates the process by which learners solve a problem, while the lower right part depicts the knowledge transformation in problem-solving. Students first comprehend the problem based on memorized knowledge and then acquire knowledge relevant to the problem. Subsequently, students apply this knowledge to find a solution and write down the answer. Finally, through feedback, they achieve knowledge internalization. Additionally, memorized knowledge will be forgotten over time.**

learners' knowledge mastery curve is relatively stable and does not fluctuate drastically in a short period. Previous work [34, 43] has noted this irrationality and mitigated the fluctuations by designing a punishment mechanism, albeit at the cost of sacrificing the model's performance.

To address the irrationality of knowledge state assessment, enhance model interpretability, and improve performance, we deconstruct students' learning processes from the perspective of human problem-solving. Engaging in exercises is a typical problem-solving process. As illustrated in Figure 1, students enhance their understanding of problems through deep thinking, continuously explore solutions, and ultimately achieve consolidation of knowledge. The entire problem-solving process involves the acquisition, application, internalization, and forgetting of memorized knowledge in the brain. However, existing KT methods overlook learners' problem-solving processes, equating memorized knowledge with acquired or applied knowledge, leading to drastic fluctuations in knowledge states. According to problem-solving theory [3, 8, 9, 27], knowledge acquisition involves learners organizing implicit and explicit knowledge relevant to problems, while knowledge application involves applying this knowledge to seek a solution. Remembering does not equal applying. Familiarity with discrete knowledge concepts (KCs) does not imply that students can effectively organize them or convert abstract knowledge into specific applications when answering problems. The common saying that *knowing the path and walking the path are two different things* reflects this phenomenon. Therefore, our work delves into various stages of problem-solving, reproducing the learning process by analyzing how students understand problems, acquire, apply, internalize, and forget knowledge.

In this paper, considering the effects of knowledge transformation mentioned above, we propose an interpretable knowledge tracing approach for problem-solving processes (PSKT). In PSKT, we first enhance the representation of knowledge-centered problems

by learning the rich attributes of the problem. Then, we present a Sequential Neural Network (SNN) to simulate knowledge transformation in problem-solving process. Before problem-solving, PSKT models students' personalized problem space and knowledge acquisition degree. During the problem-solving process, PSKT captures learners' goal-oriented knowledge application and attributes learners' response through educational parameters, namely problem factors (difficulty, discrimination), learner knowledge factors (knowledge acquisition and application), and learner behavior factors (guessing, slipping). After problem-solving, PSKT designs an update indicator to measure the degree of learners' knowledge internalization and models the forgetting effect over time. Finally, we conducted extensive experiments on five real-world datasets, demonstrating the effectiveness of PSKT. Additionally, PSKT showed outstanding performance in modeling reasonable knowledge states, explaining prediction, and learning reliable problem difficulty. The main contributions of our paper are summarized as follows:

- We quantify the learner's knowledge transformation by deconstructing the process of solving a particular problem, ensuring that knowledge state changing is reasonableness.
- We propose a novel Sequential Neural Network (SNN). The design of the SNN follows the problem-solving theory and the active construction principle of constructivist learning theory, providing good interpretability.
- Extensive experiments demonstrate the effectiveness of PSKT. In addition, PSKT can learn credible problem difficulty.

## 2 RELATED WORK

### 2.1 Item Response Theory

Item Response Theory (IRT) is a classic measurement theory rooted in education and psychology. It calculates the probability of a correct response based on the latent traits of the student and the

characteristics of the problem. Although the traditional IRT model is highly interpretable, it treats students' latent traits as a set of static parameters, making it suitable only for a single assessment scenario. Recognizing the limitations of IRT, many researchers have incorporated it into KT for process assessment. [18] integrates the IRT and BKT models to model students' learning abilities. Deep-IRT[42] fits student ability through DKVMN [45] and uses IRT to compute the probability of a student's correct response. AKT [11] uses IRT to enhance the personalized embedding of questions and responses, while PKT [34] applies IRT to fit students' knowledge mastery states.

PSKT is inspired by IRT and attributes answering performance to objective problem factors (difficulty, discrimination) and subjective learner factors (knowledge application, guessing and slipping). These rich educational parameters make our model interpretable while maintaining accuracy.

## 2.2 Knowledge Tracing

Knowledge tracing tracks students' dynamic knowledge states by modeling their historical learning sequences. It is suitable for multiple assessment scenarios and performs process measurements on students. Early KT methods were based on hidden Markov models, which evaluated student performance by modeling four parameters: initial mastery, knowledge mastery transformation, guessing and slipping [6, 29, 44]. Neural networks have greatly improved the performance of KT and have become the current mainstream method. DKT [31] introduced deep learning into KT for the first time and applied RNN/LSTM to model students' knowledge changes. Subsequent work [21, 25, 43] improved upon this basis. [1, 42, 45] use memory networks to store potential KCs and update students' knowledge mastery. [11, 16, 28] use attention mechanism to model the relationship between interaction histories, and [26, 36, 40] apply graph neural networks to model the graph structure between KCs.

Recent research has emphasized the importance of modeling the learning process. LPKT [33] and FKT [15] finely model learning and forgetting by incorporating response time and interval time. LBKT [39] highlighted the impact of answering speed, prompts, and other behaviors. However, these models suffer from poor interpretability and controllability. In contrast, our primary focus is on providing reasonable predictive explanations from an educational perspective. We disaggregate students' problem-solving processes to make multifaceted attributions of answering responses, rather than focusing on additional features of the learning process.

## 3 PROBLEM DEFINITION

Suppose that in an online learning platform, there is a set of students $S = \{s_1, s_2, ..., s_{|S|}\}$, a set of problems $P = \{p_1, p_2, ..., p_{|P|}\}$, and a set of knowledge concepts $K = \{k_1, k_2, ..., k_{|K|}\}$. We denote the exercise records of a student as $X_N = \{X_1, X_2, ..., X_N\}$, where $N$ represents the total number of exercises. $X_n = (ts_n, p_n, k_n, r_n), n \in N$ is the most basic practice unit, denoting that the student's response to problem $p_n$ at timestamp $ts_n$ is $r_n$. Where $r_n \in \{0, 1\}$, 0 denotes an incorrect answer and 1 denotes a correct answer.

**Problem Definition**. Given a student's historical learning sequence $X_n$, the task of KT is to evaluate the student's knowledge

states based on $X_n$ and predict the probability of correctly answering the next problem, that is, $p(r_{n+1} = 1|p_{n+1}, X_n)$.

## 4 THE PSKT MODEL

Students increase their knowledge by practicing problems, so the core of PSKT is quantifying learners' knowledge growth by deconstructing their problem-solving process. As shown in Figure 2, to model the learners' problem-solving process and analyze their knowledge transformation, we designed a sequence neural network (SNN). The PSKT model comprises four main modules: problem representation, knowledge acquisition, application, and updating. First, we present a knowledge-centered problem representation that enhances its expression by adjusting problem variability. Then, we model learners' personalized problem perception and knowledge acquisition degree in the knowledge acquisition module. Next, PSKT measures the knowledge application level based on knowledge acquisition and calculates problem response by IRT. Finally, the knowledge update module quantifies the degree of knowledge internalization and forgetting. The design of the SNN is inspired by problem-solving theory [9] and constructivist learning theory [10], ensuring good interpretability.

## 4.1 Problem Representation

Understanding the nature, scope, and relevant factors of a problem is prerequisite for solving it. Early KT methods used KCs to replace problems, or directly concatenated problems and KCs. However, these methods simply fused information without considering the primary and secondary relationships of the problems. Since the goal of exercises is to learn knowledge, the KCs examined by the problems are of utmost importance. Moreover, problems that test the same KCs can have different effects on learning and performance due to differences in difficulty and discrimination. In the PSKT, we consider both the primary-secondary relationships and problem attributes to enhance problem representation:

$$pa_n = \sigma\left(\mathbf{W}_{pa}^T \mathbf{P}_n + \mathbf{b}_{pa}\right), pa_n \in \{d_n, \alpha_n\}, \tag{1}$$

$$\widetilde{\mathbf{P}_n} = \mathbf{K}_n + (\phi \times d_n + \varphi \times \alpha_n) \times \mathbf{P}_n, \tag{2}$$

where $\mathbf{W}_{pa} \in \mathbb{R}^{d_h \times 1}$, $d_h$ is embedding dimension. $d_n$ is difficulty, $\alpha_n$ is discrimination, and both are scalars. The representation is inspired by $mean \pm \mu \cdot std$. $\mathbf{K}_n$ is the KC embedding, $\mathbf{P}_n$ is the original problem embedding. $(\phi \times d_n + \varphi \times \alpha_n)$ is coefficient of variation, $\phi, \varphi$ are trainable weights.

## 4.2 Knowledge Acquisition

According to problem-solving theory, knowledge acquisition requires learners to first clarify the problem, ensure an understanding of critical aspects of the problem (problem perception), and then acquire knowledge about the problem (knowledge acquisition).

**(1) Problem Perception.** The learner's solution to a problem begins with the construction of an internal representation of the external problem statement, i.e., the "problem space" (including the initial state, the goal state, and the adapted operators) [14]. To simulate the learner's personalized perception of the problem, we integrate the learner's prior knowledge $\mathbf{H}_{n-1}$ and problem representation $\widetilde{\mathbf{P}_n}$ to represent the initial state and goal state in the

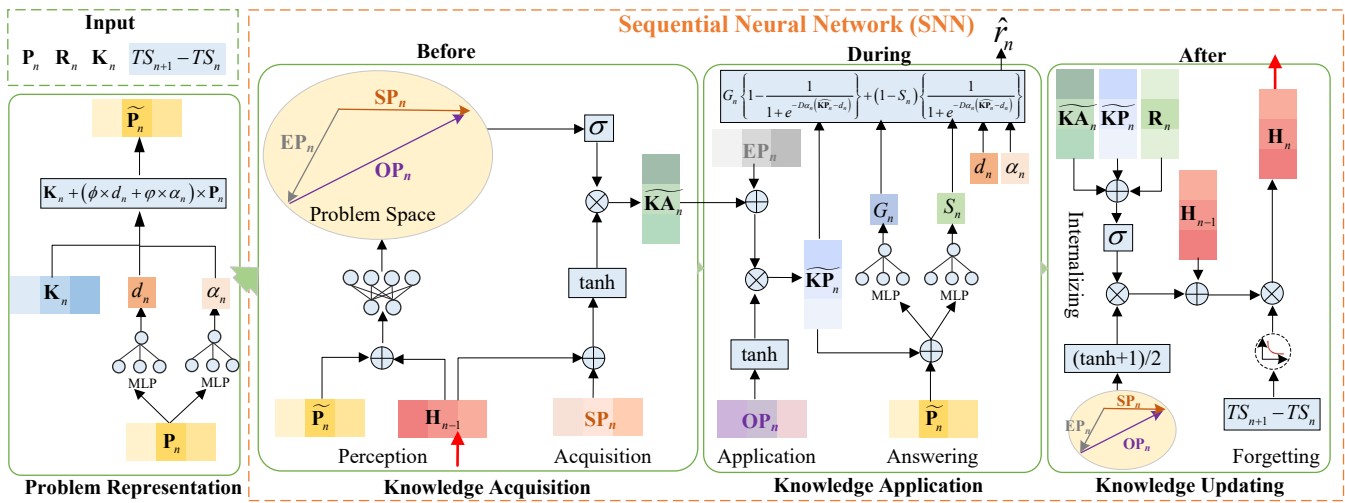

**Figure 2: The architecture of the PSKT model.**

problem space. We use the difference between the goal state $\mathbf{EP}_n$ and the initial state $\mathbf{SP}_n$ to represent the applicable operations $\mathbf{OP}_n$:

$$\mathbf{PP}_n = \mathbf{W}_{PP}^T \left( \widetilde{\mathbf{P}_n} \oplus \mathbf{H}_{n-1} \right) + \mathbf{b}_{PP}, \mathbf{PP}_n \in \{ \mathbf{SP}_n, \mathbf{EP}_n \}, \quad (3)$$

$$\mathbf{OP}_n = \mathbf{EP}_n - \mathbf{SP}_n, \quad (4)$$

where $\oplus$ is concatenate, $\mathbf{W}_{PP} \in \mathbb{R}^{(d_h+|K|) \times |K|}$ and $|K|$ is the total number of KCs.

**(2) Knowledge Acquisition.** Knowledge acquisition involves the systematic generation and integration of information. This includes acquiring knowledge from the external environment as well as constructing new knowledge through internal thought processes. In the KT scenario, external knowledge comes from the initial state of the problem $\mathbf{SP}_n$ (known conditions), and internal knowledge is derived from the learner's previous knowledge state $\mathbf{H}_{n-1}$. We use the tanh to generate candidate values. The specific formula is:

$$\mathbf{KA}_n = \tanh \left( \mathbf{W}_{KA}^T \left( \mathbf{SP}_n \oplus \mathbf{H}_{n-1} \right) + \mathbf{b}_{KA} \right), \quad (5)$$

since learners have limited memory capacity, they will selectively focus on the most relevant and core aspects of the problem during the knowledge acquisition process in order to conduct more effective planning [8]. Therefore, we further design a gate to select and retain the essential parts through the problem information.

$$\Gamma_n^{KA} = \sigma \left( \mathbf{W}_{\Gamma KA}^T \left( \mathbf{SP}_n \oplus \mathbf{EP}_n \oplus \mathbf{OP}_n \right) + \mathbf{b}_{\Gamma KA} \right), \quad (6)$$

$$\widetilde{\mathbf{KA}_n} = \Gamma_n^{KA} \times \mathbf{KA}_n, \quad (7)$$

where $\sigma$ is sigmoid, $\mathbf{W}_{KA} \in \mathbb{R}^{2|K| \times |K|}$, $\mathbf{W}_{\Gamma KA} \in \mathbb{R}^{3|K| \times |K|}$.

## 4.3 Knowledge Application

Acquiring knowledge is only the beginning of problem-solving. What is more important is how to apply knowledge to actual problems or situations. During practice, learners transform acquired theoretical knowledge into practical abilities, propose solutions (knowledge application), and then implement actions to achieve the expected goals (problem answering).

**(1) Knowledge Application.** Knowledge application involves transforming abstract knowledge into concrete action plans and adapting and refining these in practice. The solution is contained in the adapted operators of problem space, and we use tanh to generate possible solutions. Learners need to flexibly apply knowledge to make a series of dynamic decisions and continuously monitor the consequences of these decisions to achieve the goal as much as possible. Therefore, we designed a gate $\Gamma^{KP}$ led by problem goals and knowledge acquisition to determine the learner's final solution.

$$\Gamma_n^{KP} = \sigma \left( \mathbf{W}_{\Gamma KP}^T \left( \mathbf{EP}_n \oplus \widetilde{\mathbf{KA}_n} \right) + \mathbf{b}_{\Gamma KP} \right), \quad (8)$$

$$\widetilde{\mathbf{KP}_n} = \Gamma_n^{KP} \times \tanh \left( \mathbf{OP}_n \right). \quad (9)$$

**(2) Problem Answering.** Applying knowledge is to achieve goals. IRT shows that whether students can answer problems correctly depends not only on their knowledge application but also on the problem itself. Inspired by the Four-parameter IRT and BKT, we simulate students' responses by fully considering problem factors (difficulty, discrimination), learner knowledge factors (knowledge application), and learner behavioral factors (guessing and slipping). The core functions are as follows:

$$\widehat{\mathbf{KP}_n} = \sigma \left( \mathbf{W}_2^T \cdot \text{ReLU} \left( \mathbf{W}_1^T \left( \widetilde{\mathbf{KP}_n} \oplus \widetilde{\mathbf{P}_n} \right) + \mathbf{b}_1 \right) + \mathbf{b}_2 \right) \quad (10)$$

$$\hat{r}_n = G_n \left\{ 1 - \frac{1}{1 + e^{-D\alpha_n \left( \widehat{\mathbf{KP}_n} - d_n \right)}} \right\} + (1 - S_n) \left\{ \frac{1}{1 + e^{-D\alpha_n \left( \widehat{\mathbf{KP}_n} - d_n \right)}} \right\}, \quad (11)$$

where $\mathbf{W}_1 \in \mathbb{R}^{(|K|+d_h) \times d_h}$, $\mathbf{W}_2 \in \mathbb{R}^{d_h \times |K|}$. The first term in Equation (11) represents the probability that the learner has not mastered the knowledge but guessed correctly, and the second term indicates that the learner can answer correctly without slipping. $D$ is a constant with a value of $4 \times 1.7$ [4]. $G_n$ is guessing, and $S_n$ is slipping:

$$GS_n = \lambda \times \sigma \left( \mathbf{W}_{GS}^T \left( \widehat{\mathbf{KP}_n} \oplus \widetilde{\mathbf{P}_n} \right) + \mathbf{b}_{GS} \right), GS_n \in \{ G_n, S_n \}, \quad (12)$$

where $\lambda$ is hyperparameter, indicating the threshold of guessing and slipping, $\mathbf{W}_{GS} \in \mathbb{R}^{(|K|+d_h) \times 1}$.

## 4.4 Knowledge Updating

After practice, the learner's knowledge state changes. Constructivist learning theory [10] suggests that increasing knowledge mastery reflects knowledge internalization, while forgetting leads to a reduction in knowledge. Thus, learner state updating includes both knowledge internalization and forgetting.

**(1) Knowledge internalizing.** Knowledge internalizing is how learners transform external knowledge, experience or information into thinking structures and cognitive patterns. In the problem-solving process, the learner's new knowledge originates from the problems, so we fuse all the information of the problems to obtain the candidate value of knowledge increment. Constructivist learning theory shows that learning is an active construction process, so we tend to think that the entire practice process positively impacts learners, and use $(\tanh(.) + 1)/2$ as a non-negative constraint operator to ensure that the increment is non-negative.

$$\mathbf{KI}_n = \left(\tanh\left(\mathbf{W}_{KI}^T\left(\mathbf{SP}_n \oplus \mathbf{EP}_n \oplus \mathbf{OP}_n\right) + \mathbf{b}_{KI}\right) + 1\right)/2. \quad (13)$$

Through internalization, learners do not simply accept information but integrate it into their cognitive framework. Thus, even if different learners answer the same problem, the level of knowledge they internalize will be different due to differences in the knowledge they acquire, the knowledge they apply, and the results of their answers. To this end, we propose a growth indicator to determine the learner's personalized knowledge increment.

$$\Gamma_n^{KI} = \sigma\left(\mathbf{W}_{\Gamma KI}^T\left(\widetilde{\mathbf{KA}_n} \oplus \widetilde{\mathbf{KP}_n} \oplus \mathbf{R}_n\right) + \mathbf{b}_{\Gamma KI}\right), \quad (14)$$

$$\widetilde{\mathbf{H}_n} = \mathbf{H}_{n-1} + \Gamma_n^{KI} \times \mathbf{KI}_n, \quad (15)$$

where $\mathbf{R}_n$ is response embedding, $\mathbf{W}_{\Gamma KI} \in \mathbb{R}^{(2|K|+d_h)\times|K|}$.

**(2) Knowledge Forgetting.** Limited by the human memory system, although learners may delay the occurrence of forgetting through review and reinforcement learning, forgetting is still a continuous process [22]. Therefore, current knowledge may be forgotten over time intervals. In order to model the complex forgetting effect, we design a forgetting gate as follows:

$$\Gamma_n^{KF} = \sigma\left(\mathbf{W}_{\Gamma KF}^T\left(\mathbf{IT}_n \oplus \widetilde{\mathbf{P}_{n+1}} \oplus \widetilde{\mathbf{H}_n}\right) + \mathbf{b}_{\Gamma KF}\right), \quad (16)$$

$$\mathbf{H}_n = \Gamma_n^{KF} \times \widetilde{\mathbf{H}_n}, \quad (17)$$

where time interval is $it_n = ts_{n+1} - ts_n$, in minutes. $\mathbf{IT}_n$ is embedding of $it_n$, and we set all intervals greater than 1 month to 1 month.

## 4.5 Objective Function

To train all parameters $\Theta$ in PSKT, we use cross-entropy log loss between the predicted response $\hat{r}_n$ and the actual response $r_n$ as the objective function, minimized with the Adam optimizer:

$$\mathcal{L}_r(\Theta) = \sum_{n=1}^{N} -\left[r_n \log \hat{r}_n + (1 - r_n)\log(1 - \hat{r}_n)\right], \quad (18)$$

## 5 EXPERIMENTS

In this section, we conduct extensive experiments to answer the following questions:

- **RQ1.** What is the predictive performance of PSKT?
- **RQ2.** Can PSKT reasonably explain the learning process?

**Table 1: Details of the all datasets.**

| datasets | Students | Problems | KCs | Responses |
|---|---|---|---|---|
| ASSIST12 | 29,018 | 53,091 | 265 | 2,711,813 |
| ASSIST17 | 1,709 | 3,162 | 102 | 942,816 |
| EdNet-KT1 | 100,000 | 12,267 | 189 | 12,092,643 |
| Junyi | 247,606 | 722 | 41 | 25,926,003 |
| Eedi | 118,971 | 27,613 | 1,989 | 15,867,850 |

- **RQ3.** How effective is the problem representation of PSKT?
- **RQ4.** What is the impact of each element in PSKT?

### 5.1 Experimental Setting

*5.1.1 Dataset.* We selected five commonly used and large public datasets in KT to evaluate the effectiveness of PSKT: ASSIST12[1], ASSIST17[2], EdNet-KT1[3], Junyi[4], Eedi[5]. The statistical information of datasets is shown in Table 1. We filtered learning records for all datasets with missing relevant KCs and removed problems with fewer than 10 occurrences and learners with fewer than 3 response records. Due to the large size of the Ednet-KT1, which contains 95,293,926 records, we randomly selected 100,000 learners [34, 41]. For datasets where a problem assesses multiple KCs, we use the average of all KC embeddings to represent the final KC embedding.

*5.1.2 Training Details.* We set all input sequences to a fixed length of 100. For sequences of length greater than 100, we cut them into several unique subsequences. We performed the 5-fold cross-validation. For each fold, 80% of the learned sequences are split into training and validation sets (the ratio is 8:2), and the remaining 20% serves as the test set. The parameters $d_h$, learning rate, $\lambda$ are set to 256, 0.001, 0.5. On the smaller ASSIST12 and ASSIST17 datasets, the batch size is 64, while on other larger datasets is 512. We used an early stopping mechanism with 3 epochs of patience. For fairness, the baselines' hyper-parameters are carefully tuned to achieve optimal performance. All experiments were performed on a Linux server with an NVIDIA GeForce RTX3090.

*5.1.3 Baselines.* We compared PSKT with 11 baselines to evaluate its effectiveness.

- **DKT** uses the hidden state of RNN to represent the learner's knowledge state. [31]
- **DKVMN** defines a static KC matrix and a dynamic state matrix, using read and write operations to update state. [45]
- **SAKT** utilizes the self-attention mechanism to capture the relationships between historical records. [28]
- **Deep-IRT** utilizes DKVMN to estimate the learner's ability and uses IRT to compute the correct probability. [42]
- **DKT-F** incorporates temporal information into DKT to model learner forgetting behavior. [25]
- **AKT** enhances problems' personalized embedding by IRT and captures the learner's state by attention mechanism. [11]

[1]https://sites.google.com/site/assistmentsdata/2012-13-school-data-withaffect
[2]https://sites.google.com/view/assistmentsdatamining/dataset
[3]https://github.com/riiid/ednet
[4]https://pslcdatashop.web.cmu.edu/DatasetInfo?datasetId=1198
[5]https://eedi.com/projects/neurips-education-challenge

**Table 2: Comparative results of student response prediction. The best results are in bold, and the next best are underlined. * indicates p-value < 0.05 in the t-test compared to the second-best result.**

| Datasets | Metrics | DKT | DKVMN | SAKT | Deep-IRT | DKT-F | AKT | SAINT | LPKT | ATDKT | FKT | PKT | PSKT |
|---|---|---|---|---|---|---|---|---|---|---|---|---|---|
| ASSIST -12 | AUC | 0.7191 | 0.7261 | 0.7741 | 0.7660 | 0.7504 | 0.7938 | 0.7856 | 0.7974 | 0.7631 | 0.7931 | 0.7938 | **0.8252***  |
| | ACC | 0.7313 | 0.7301 | 0.7553 | 0.7522 | 0.7451 | 0.7664 | 0.7640 | 0.7692 | 0.7519 | 0.7667 | 0.7647 | **0.7838*** |
| | RMSE | 0.4272 | 0.4258 | 0.4090 | 0.4116 | 0.4169 | 0.4006 | 0.4037 | 0.3987 | 0.4120 | 0.4006 | 0.4011 | **0.3858*** |
| | R2 | 0.1354 | 0.1410 | 0.2077 | 0.1992 | 0.1786 | 0.2415 | 0.2305 | 0.2468 | 0.1976 | 0.2400 | 0.2398 | **0.2966*** |
| ASSIST -17 | AUC | 0.7295 | 0.7511 | 0.7355 | 0.7500 | 0.7518 | 0.7781 | 0.7649 | 0.8058 | 0.7488 | 0.7888 | 0.7947 | **0.8182*** |
| | ACC | 0.6940 | 0.7043 | 0.7053 | 0.7139 | 0.7054 | 0.7210 | 0.7113 | 0.7414 | 0.7026 | 0.7326 | 0.7327 | **0.7503*** |
| | RMSE | 0.4454 | 0.4379 | 0.4419 | 0.4364 | 0.4376 | 0.4308 | 0.4323 | 0.4136 | 0.4389 | 0.4224 | 0.4194 | **0.4075*** |
| | R2 | 0.1556 | 0.1839 | 0.1690 | 0.1896 | 0.1852 | 0.2101 | 0.2048 | 0.2719 | 0.1803 | 0.2408 | 0.2515 | **0.2933*** |
| EdNet -KT1 | AUC | 0.7128 | 0.7401 | 0.7590 | 0.7600 | 0.7316 | 0.7623 | 0.7509 | 0.7645 | 0.7413 | 0.7639 | 0.7588 | **0.7749*** |
| | ACC | 0.6961 | 0.7138 | 0.7213 | 0.7229 | 0.7054 | 0.7247 | 0.7166 | 0.7262 | 0.7099 | 0.7247 | 0.7216 | **0.7318*** |
| | RMSE | 0.4438 | 0.4346 | 0.4285 | 0.4280 | 0.4379 | 0.4271 | 0.4316 | 0.4251 | 0.4348 | 0.4267 | 0.4286 | **0.4220*** |
| | R2 | 0.129 | 0.1646 | 0.1882 | 0.1900 | 0.1521 | 0.1933 | 0.1763 | 0.2009 | 0.1639 | 0.1946 | 0.1876 | **0.2125*** |
| Junyi | AUC | 0.7510 | 0.7964 | 0.7976 | 0.7971 | 0.7646 | 0.7995 | 0.7958 | 0.8004 | 0.7839 | 0.8035 | 0.7970 | **0.8057*** |
| | ACC | 0.8457 | 0.8523 | 0.8531 | 0.8525 | 0.8477 | 0.8537 | 0.8527 | 0.8530 | 0.8503 | 0.8550 | 0.8539 | **0.8558*** |
| | RMSE | 0.3445 | 0.3336 | 0.3329 | 0.3333 | 0.3414 | 0.3322 | 0.3337 | 0.3323 | 0.3370 | 0.3309 | 0.3326 | **0.3296*** |
| | R2 | 0.1584 | 0.2108 | 0.2140 | 0.2121 | 0.1735 | 0.2173 | 0.2103 | 0.2168 | 0.1949 | 0.2233 | 0.2155 | **0.2298*** |
| Eedi | AUC | 0.7714 | 0.7691 | 0.7760 | 0.8068 | 0.7813 | 0.8082 | 0.8045 | 0.8053 | 0.7928 | 0.8065 | 0.8087 | **0.8201*** |
| | ACC | 0.7240 | 0.7224 | 0.7244 | 0.7470 | 0.7313 | 0.7480 | 0.7450 | 0.7461 | 0.7375 | 0.7476 | 0.7487 | **0.7571*** |
| | RMSE | 0.4273 | 0.4278 | 0.4253 | 0.4113 | 0.4228 | 0.4106 | 0.4126 | 0.4120 | 0.4179 | 0.4114 | 0.4104 | **0.4045*** |
| | R2 | 0.2042 | 0.2023 | 0.2116 | 0.2628 | 0.2210 | 0.2651 | 0.2581 | 0.2602 | 0.2389 | 0.2625 | 0.2661 | **0.2867*** |

- **SAINT** directly applies transformer technology and separates problems and responses in interactions. [5]
- **LPKT** uses response times and intervals to model learning process, calculating learning gains and forgetting. [33]
- **ATDKT** improves performance by adding KC prediction and learner prior knowledge prediction tasks. [21]
- **PKT** uses IRT to compute students' knowledge states and models answering behaviors (guessing and slipping). [34]
- **FKT** feeds response and interval time into the transformer to enrich the learner's observable performance. [15]

## 5.2 Student Performance Prediction (Q1)

We measure model performance in terms of area under the curve (AUC), accuracy (ACC), root mean square error (RMSE), and Pearson correlation squared (R2). Consistent with previous work [15, 33, 35], we set the accuracy threshold to 0.5. We used the average result of 5-fold cross-validation on the test set as the experimental result. Since the Eedi does not have response time features, the FKT and LPKT results on Eedi are after eliminating response time.

The experimental results are shown in Table 2, and PSKT outperforms all baseline methods on all datasets and metrics. The excellent performance of PSKT shows that comprehensively modeling the learner's problem-solving process significantly impacts understanding the learning process and predicting learning performance. Secondly, compared with the most advanced deep learning models, the superiority of the PSKT framework also shows that the factor that restricts the performance of the current KT method is not the expressive ability of the algorithm but the design of a scientific framework that conforms to students' cognitive rules.

Third, the psychometric model IRT uses educational parameters to enhance the interpretability of the model. Among the models combined with IRT (AKT, Deep-IRT, PKT, PSKT), PSKT performs best, further proving the PSKT algorithm mechanism. The superiority of it takes into account both high accuracy and interpretability.

## 5.3 Explain the learning process (Q2)

To prove that PSKT can capture students' reasonable knowledge states, we conducted experiments on learning process analysis. In Figure 3, we randomly select a student from each dataset and visualize their learning process when they practiced the same KC consecutively. We illustrate the superiority of PSKT in two aspects.

**(1) Knowledge state rationality.** The rationality of the knowledge state includes the stability of the knowledge state curve and the rationality of the knowledge state value. First, the knowledge state curve fitted by PSKT is more stable. We quantify fluctuations based on the differences in knowledge states for all KCs between each student's adjacent two responses. As shown in the box plots in Figure 4 (a), PSKT is the most stable across the three models. Learners' knowledge mastery is directional and will not fluctuate drastically in a short period [24]. As shown in Figure 3, the knowledge concept curves of Deep-IRT and DKT often decline and rise sharply. In contrast, PKT has more minor fluctuations because it designs a penalty mechanism for knowledge state changes. However, this mechanism can only alleviate the irrationality but does not really solve the problem and sacrifices the model's performance. On the contrary, PSKT is based on human problem-solving theory and uses a scientific and reasonable framework to make the curve more directional.

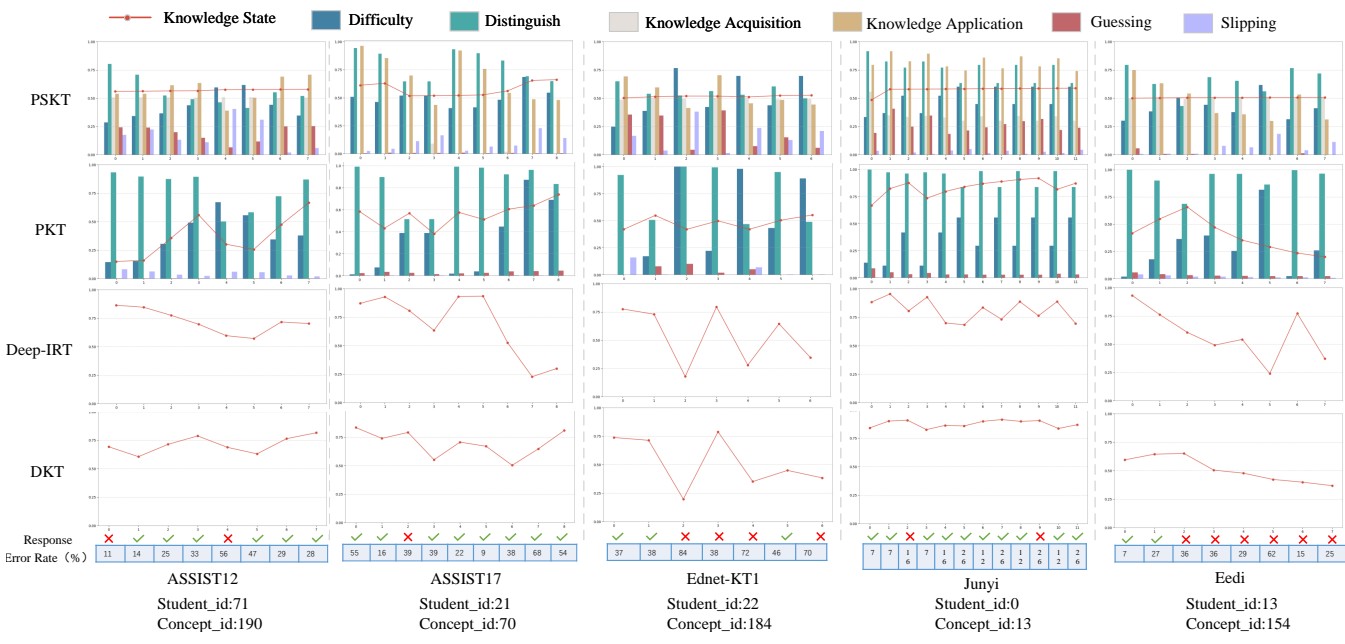

Figure 3: Compare the interpretability of DKT, Deep-IRT, PKT and PSKT for the learning process.

Secondly, PSKT tracked more reasonable knowledge state values. DKT, Deep-IRT, and PKT obtained high knowledge values (>0.5) when learners responded correctly and decreased when incorrect. In ASSIST12, when the student answered the 4th (indices start from 0) problem incorrectly, the PKT knowledge value dropped from 0.56 to 0.30 and significantly increased when he/she answered the last three problems correctly (0.31→0.71). PSKT believes learners' knowledge states have always been relatively stable (all around 0.55) and attributes correct answers to the improvement of students' application levels. First, the 4th problem was the most difficult of the whole practice (56% error rate), so the student answered incorrectly because of a mismatch between knowledge state and difficulty rather than a decline in knowledge. The reason that learners can answer the last three exercises correctly is not because their knowledge has increased significantly but because the learners' knowledge application level has improved during the practice.

**(2) Learning process interpretability.** Compared with other methods, PSKT provides more educational parameters to explain the learning process. DKT attributes student performance entirely to knowledge mastery, Deep-IRT adds the effect of problem difficulty, and PKT considers student guessing and slipping behavior. However, these models rely heavily on a sudden drop in knowledge mastery to explain responses incorrectly, whereas PSKT provides a more reasonable explanation. For example, on the Eedi dataset, the student incorrectly answered six consecutive problems that examined 154 KC, which DKT and PKT attributed to a decrease in students' knowledge mastery. We think that the practice process positively affects learners and that students' knowledge mastery does not decline significantly during continuous answering. Specifically, although students answered problems 1 and 2 correctly, PSKT considered that learners' mastery of this knowledge was still only about 0.5 because of the low difficulty of these two problems. The

student answered the last 6 problems incorrectly, though, which PSKT attributed to the high difficulty and the low level of knowledge application. At the same time, students correctly answered the first problem (more difficult) but incorrectly answered the last two problems, which PSKT attributed to the slipping.

## 5.4 Study the Problem Representation(Q3)

In this section, we conduct some experiments to demonstrate the reliability of the problem representation learned in PSKT, including the effectiveness of problem difficulty assessment and the interpretability of the problem representation.

**(1) Validity of problem difficulty assessment.** Since there is no problem difficulty indicator in all datasets, to evaluate the effectiveness of the difficulty learned by PSKT, we refer to the Classical Test Theory [37] and previous KT work that considers difficulty [24, 32, 46], and defines the statistical difficulty of the problem as the error rate. We conducted a Pearson correlation test between the two difficulties. The experimental objects are the problems involved in the test set. The experimental results are shown in subgraph (b1) in Figure 4. PSKT learned difficulty is strongly correlated with statistical difficulty (>0.7) on all datasets, which demonstrates the reliability of its problem difficulty. In addition, the scatter plot on ASSIST17 also shows a certain degree of correlation and consistency between the two difficulties (Figure 4 (b2)).

**(2) Interpretability of problem representation.** We visualize the correlation weights between problems by computing the cosine similarity of the problem representations, as shown in Figure 5. We randomly selected eight questions on ASSIST17. These problems tested two KCs (9, 74). First, the correlation weights tend to be higher between problems that test the same KC, such as problems 1, 2, and 3. Secondly, when different problems have the same KC,

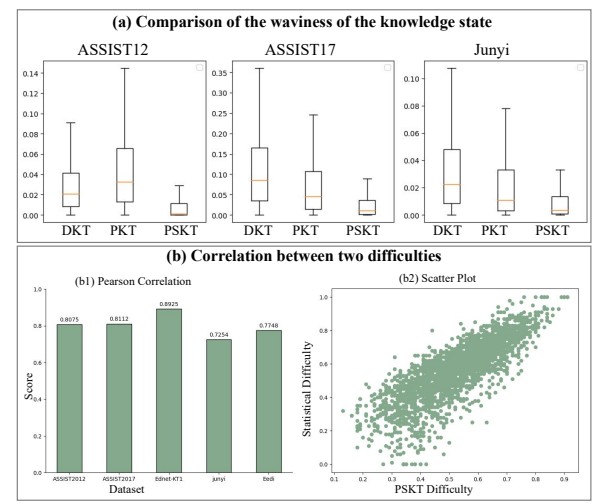

**Figure 4: Visualization of knowledge state fluctuations and correlation between two difficulties.**

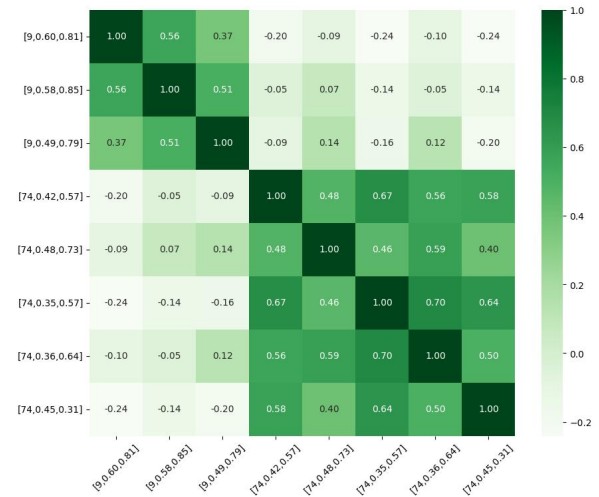

**Figure 5: Weights of problem representations on ASSIST17. The label $[i, j, k]$ indicates $[KC, difficulty, discrimination]$.**

the closer the difficulty and discrimination of the problems are, the higher the similarity, for example, [9,0.60,0.81] and [9,0.58,0.85].

## 5.5 Ablation experiments(Q4)

We explore the impact of each module in PSKT through ablation experiments. As shown in Table 3, we selected 6 variants, each removing an element from the PSKT. PR is replacing the problem representation with concatenate, EP denotes removing problem perception, KA means removing knowledge application, and KP denotes removing knowledge acquisition. KI is knowledge internalization without considering knowledge acquisition and Knowledge application, KF represents removing knowledge forgetting.

First, knowledge forgetting is vital in the learning process, and deleting this module causes the most significant performance drop.

**Table 3: Results of ablation experiments.**

| Data | Metrics | PR | EP | KA | KP | KI | KF | PSKT |
|---|---|---|---|---|---|---|---|---|
| ASSIST-12 | AUC | 0.8248 | 0.8158 | 0.8243 | 0.8232 | 0.8244 | 0.8132 | **0.8252** |
| | ACC | 0.7834 | 0.7784 | 0.7823 | 0.7817 | 0.7830 | 0.7764 | **0.7838** |
| | RMSE | 0.3861 | 0.3905 | 0.3866 | 0.3871 | 0.3864 | 0.3918 | **0.3858** |
| | R2 | 0.2954 | 0.2793 | 0.2935 | 0.2918 | 0.2942 | 0.2744 | **0.2966** |
| ASSIST-17 | AUC | 0.8175 | 0.8146 | 0.8157 | 0.8166 | 0.8141 | 0.8078 | **0.8182** |
| | ACC | 0.7496 | 0.7476 | 0.7485 | 0.7490 | 0.7473 | 0.7433 | **0.7503** |
| | RMSE | 0.4080 | 0.4096 | 0.4088 | 0.4083 | 0.4097 | 0.4127 | **0.4075** |
| | R2 | 0.2917 | 0.2861 | 0.2889 | 0.2905 | 0.2856 | 0.2753 | **0.2933** |
| Ednet-KT1 | AUC | 0.7745 | 0.7729 | 0.7686 | 0.7724 | 0.7667 | 0.7640 | **0.7749** |
| | ACC | 0.7311 | 0.7329 | 0.7277 | 0.7304 | 0.7261 | 0.7247 | **0.7318** |
| | RMSE | 0.4222 | 0.4223 | 0.4247 | 0.4230 | 0.4256 | 0.4264 | **0.4220** |
| | R2 | 0.2119 | 0.2151 | 0.2025 | 0.2089 | 0.1991 | 0.1961 | **0.2125** |
| Junyi | AUC | 0.8052 | 0.8042 | 0.8048 | 0.8049 | 0.8042 | 0.8040 | **0.8057** |
| | ACC | 0.8555 | 0.8550 | 0.8553 | 0.8554 | 0.8552 | 0.8555 | **0.8558** |
| | RMSE | 0.3298 | 0.3303 | 0.3300 | 0.3299 | 0.3302 | 0.3303 | **0.3296** |
| | R2 | 0.2286 | 0.2262 | 0.2280 | 0.2281 | 0.2267 | 0.2265 | **0.2298** |
| Eedi | AUC | 0.8194 | 0.8193 | 0.8183 | 0.8180 | 0.8170 | 0.8090 | **0.8201** |
| | ACC | 0.7569 | 0.757 | 0.7561 | 0.7555 | 0.7547 | 0.7482 | **0.7571** |
| | RMSE | 0.4047 | 0.4049 | 0.4054 | 0.4057 | 0.4065 | 0.4103 | **0.4045** |
| | R2 | 0.2861 | 0.2854 | 0.2839 | 0.2827 | 0.2799 | 0.2663 | **0.2867** |

According to the design of PSKT, if forgetting is not considered, the learner's knowledge mastery will continue to increase (or remain unchanged) as the number of exercises increases, making it challenging to identify incorrect answers. Secondly, the impact of problem perception is also significant. This is because problem perception involves modeling the problem's initial state, the target state, and the adapted operator. These three variables are throughout the entire problem-solving process. Third, both knowledge acquisition and knowledge application contribute to the performance of PSKT, proving that knowledge acquisition and knowledge application are both indispensable processes for problem-solving.

## 6 CONCLUSION

In this paper, we propose a problem-solving process-oriented knowledge tracing model (PSKT) to explore the cognitive process of how students transform theoretical knowledge into practical skills. According to the three stages of practice, before problem-solving, we first obtain students' personalized perceptions of the problem, and simulate the students' acquisition of the problem-related knowledge. During problem-solving, we model the learner's level of knowledge application toward the problem target and use educationally meaningful parameters for performance attribution to predict the learner's response. After problem-solving, we design an update indicator to measure their level of knowledge internalization and the forgetting effect over time intervals. Finally, we conducted extensive experiments to validate the effectiveness of PSKT. In addition, PSKT not only performs well in response prediction, but also more reasonably evaluates learners' knowledge mastery patterns, provides educational insights into students' success or failure in problems. Looking forward, PSKT still has room for improvement. For example, we can consider collecting the details of students' answers to problems and using large language model technology to capture students' knowledge transformation in a more fine-grained manner from the students' answer steps.

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
