# OpenReview forum: "Remembering is Not Applying: Interpretable Knowledge Tracing for Problem-solving Processes"
_acmmm.org/ACMMM/2024/Conference — MM2024 Poster_

### Official Review · Reviewer_CbND · 2024-05-01

**Rating:** 4
**Confidence:** 3

**Summary:**

The paper presents a significant advancement in the field of knowledge tracing by addressing the application of knowledge in problem-solving contexts. However, it would benefit from addressing the following weaknesses, particularly enhancing the discussion on practical implementation challenges and model generalizability. Enhancing the clarity and depth of theoretical motivations and providing a more comparative analysis of computational efficiency would also strengthen the paper.

**Strengths:**

1.	Interpretable Model: The PSKT model offers high interpretability, enabling predictions of student performance and simulations of knowledge transformation during problem-solving.
2.	Problem-Solving Focus: It emphasizes problem-solving processes, enhancing its application in distance education and adaptive learning systems.
3.	Educational Parameters: PSKT utilizes educationally relevant parameters to assess learner knowledge states, promoting a deeper understanding of the learning process.
4.	Comprehensive Framework: This model incorporates multiple stages of learning, including knowledge acquisition, application, internalization, and forgetting, offering a comprehensive view.
5.	Performance: It outperforms existing knowledge transformation (KT) methods in various datasets, showcasing its predictive accuracy and effectiveness in assessing knowledge states.
6.	Extensive Evaluation: Validated through rigorous testing on five real-world datasets, demonstrating its robustness and effectiveness.
7.	Educational Insights: Provides valuable insights into students' learning processes, aiding educators in refining teaching strategies.

**Limitations:**

1.	Challenges with the Problem-Solving Model: The paper highlights that the interpretability of problem-solving knowledge tracing (PSKT) models still faces challenges. While the models strive to interpret knowledge states and problem-solving processes, the precise mechanisms by which knowledge is transformed during problem-solving activities are not thoroughly explained or understood. This leaves a gap in how accurately the model can interpret complex learning and application processes.
2.	Modeling Knowledge Application: The paper emphasizes that current knowledge tracing methods, including PSKT, primarily focus on modeling knowledge application as a uniform process without considering the variability in how individuals apply knowledge in diverse contexts. This could lead to oversimplifications in the models, potentially misrepresenting the ways students apply learned concepts to solve problems.
3.	Need for Enhanced Data Collection: There is a suggestion for improvement in data collection methods to capture finer details of students’ problem-solving processes. The current models may not effectively capture all variations in how knowledge is applied, which is critical for improving the accuracy and utility of knowledge tracing systems.
4.	Potential for Improvement: The authors suggest that there is room for enhancing the model by collecting more detailed data on students’ responses to problems and integrating large language model technology. This indicates a recognition of the current limitations in capturing fine-grained details of students' knowledge transformation during problem-solving processes.
5.	The introduction and literature review sections could be expanded to better situate the research within the broader context of adaptive learning technologies and educational data mining.
6.	Algorithm Transparency and Interpretability: Despite the PSKT model's emphasis on interpretability, the explanations of the model's internal decision processes remain somewhat abstract. For instance, a clearer explanation of how each parameter of the model elucidates cognitive and behavioral changes during the learning process and their direct applications in educational practice could be more explicitly stated.
7.	Logical Flow and Organization: Some sections require stronger structure and logical coherence. Especially in the methodology and results sections, the way information is organized can sometimes seem disjointed, potentially hindering the reader's understanding of the entire text.
8.	Consistency in Terminology Use: The article needs more consistent definitions and use of key concepts such as "knowledge acquisition," "knowledge application," and "problem-solving." Clearer definitions could prevent confusion.

**Suitability:**

2

---

### Official Review · Reviewer_RXsW · 2024-05-24

**Rating:** 5
**Confidence:** 4

**Summary:**

This paper introduces the Problem-solving Knowledge Tracing (PSKT) model that adeptly captures the dynamic knowledge state of students across three distinct stages: before, during and after problem-solving, thus offering a comprehensive framework for understanding the learning process. By integrating insights from constructivist learning theories and incorporating the Rasch model, the representation of the question and the interpretability of knowledge application stage are consequently enhanced. The experimental results on several public datasets demonstrates the effectiveness of the PSKT model.

**Strengths:**

1. The authors investigate the constructivist learning theories and Rasch model to enhance the PSKT’s interpretability.
2. The paper conduct experiments and analyses on the interpretability of the PSKT model including the stability of the knowledge state and the correlation between questions’ representation.

**Limitations:**

I have the following concerns and I hope the authors could make some further illustrations:
Q1: As LPKT is specifically focusing on the modelling of learning process, it’s recommended to compare the knowledge state rationality and interpretability with LPKT.
Q2: Since the authors have validated the interpretability of problem factors, e.g. assessing the generated problem difficulty’s correlation with the statistical difficulty. I hope the authors could also investigate the interpretability of the learner knowledge factors, i.e., guessing and slipping factor. This extension could involve exploring whether there exist discernible insights into potential significant differences in the guessing and slipping factors among students at varying levels of knowledge mastery.
Q3: In Figure 3, the consistent centrality of the knowledge state of PSKT around 0.5, irrespective of whether the majority of questions are answered correctly or incorrectly, with minimal fluctuations, appears unreasonable. The provided explanation is not very convincing, and it is advisable for the authors to provide a more in-depth analysis to elucidate this observation.

**Suitability:**

2

---

### Official Review · Reviewer_so9s · 2024-05-28

**Rating:** 3
**Confidence:** 4

**Summary:**

This paper focuses on knowledge transformation in the problem-solving process and proposes an interpretable Problem-solving Knowledge Tracing (PSKT) method. Unlike other studies, the authors first propose a knowledge-centered representation of the problem, which is enhanced by adjusting the variability of the problem. Meanwhile, the authors elaborated a Sequential Neural Network (SNN) in three stages (before problem-solving, during problem-solving, and after problem-solving).

**Strengths:**

1.PSKT exhibits outstanding performance compared to existing knowledge tracing models.
2.The authors conducted sufficient experiments and provided in-depth analysis of the results.
3.The manuscript is well-written, with clear logic, a rational experimental design.
4.The figures are well-drawn and contribute to a better understanding of the model.
5. Problem solving theory as an entry point is interesting.

**Limitations:**

1. This paper starts with the problem-solving theory to construct the model. Can you explain in detail how each component of the PSKT model fits with the problem-solving theory? You can list the corresponding theoretical basis. If you can answer my question, I am willing to give you a higher score.
2. Figure 2 could benefit from some legend explanations to clarify the meaning, as excessive letters and symbols make it difficult to discern. The inputs and outputs of each module are not very apparent.
3. As far as I know, there are many KT model studies on learning processes. What are the significant differences between the model design proposed by the author and them?
4. The font in Figures 3 and 4 is too small, and there is no necessary textual description, which makes it difficult for readers to understand the meaning conveyed by these figures. For example, what do "Knowledge Acquisition" and "Knowledge Application" mean in Figure 3?
5. The author's motivation primarily mentions that PSKT focuses on the interpretability of the model. As far as I know, the visualization of knowledge state evolution, including the visualization of problem difficulty and discrimination, has been addressed in most KT articles, and the novelty of interpretability related to problem-solving theory has not been observed.

**Suitability:**

2

---

### Meta-Review · Area_Chair_EXLs · 2024-07-06

**Recommendation:** Accept (Poster)
**Confidence:** 4

**Metareview:**

All the reviewers acknowledged this paper's significant contribution and improved performance in knowledge tracing within problem-solving contexts. A common concern among the reviewers was the lack of clarity on transparency and interpretability. Upon reviewing both the paper and the authors' responses, it is clear that the paper introduces fresh ideas—integrating insights from constructivist learning theories and incorporating the Rasch model—and is well written. The authors have satisfactorily addressed the reviewers' concerns within the space limitations. One reviewer did not enter a final rating but acknowledged checking the rebuttal and decided to maintain their score of borderline accept.